# Source Apportionment and Toxicity of PM in Urban, Sub-Urban, and Rural Air Quality Network Stations in Catalonia

**Clara Jaén** [1] (ID), **Paula Villasclaras** [1], **Pilar Fernández** [1] (ID), **Joan O. Grimalt** [1] (ID), **Mireia Udina** [2] (ID), **Carmen Bedia** [1] (ID) **and Barend L. van Drooge** [1,*] (ID)

1   Institute of Environmental Assessment and Water Research (IDAEA-CSIC), c/Jordi Girona 18-26, 08034 Barcelona, Spain; clara.jaen@idaea.csic.es (C.J.); pvillaca27@alumnes.ub.edu (P.V.); pilar.fernandez@idaea.csic.es (P.F.); joan.grimalt@idaea.csic.es (J.O.G.); carmen.bedia@idaea.csic.es (C.B.)
2   Department of Applied Physics—Meteorology, University of Barcelona, C/ Martí i, Franquès, 1, 08028 Barcelona, Spain; mudina@meteo.ub.edu
*   Correspondence: barend.vandrooge@idaea.csic.es

**Abstract:** Air quality indicators, i.e., $PM_{10}$, $NO_2$, $O_3$, benzo[a]pyrene, and several organic tracer compounds were evaluated in an urban traffic station, a sub-urban background station, and a rural background station of the air quality network in Catalonia (Spain) from summer to winter 2019. The main sources that contribute to the organic aerosol and PM toxicity were determined. Traffic-related air pollution dominated the air quality in the urban traffic station, while biomass burning in winter and secondary organic aerosol (SOA) in summer impact the air quality in the sub-urban and rural background stations. Health risk assessment for chronic exposure over the past decade, using WHO air quality standards, showed that $NO_2$, $PM_{10}$ and benzo[a]pyrene from traffic emissions pose an unacceptable risk to the human population in the urban traffic station. $PM_{10}$ and benzo[a]pyrene from biomass burning were unacceptably high in the sub-urban and rural background stations. Toxicity tests of the PM extracts with epithelial lung cells showed higher toxicity in wintertime samples in the sub-urban and rural stations, compared to the urban traffic station. These results require different mitigation strategies for urban and rural sites in order to improve the air quality. In urban areas, traffic emissions are still dominating the air quality, despite improvements in the last years, and may directly be responsible for part of the SOA and $O_3$ levels in sub-urban and rural areas. In these later areas, air pollution from local biomass burning emissions are dominating the air quality, essentially in the colder period of the year.

**Keywords:** particulate matter; organic aerosol; toxicity; source apportionment; health risk assessment

## 1. Introduction

Air pollution is a global threat to ecosystems and affects human health, even at short-term exposure [1,2]. Therefore, there is a growing demand to improve the air quality [3–5]. In Europe, the European Commission supports the Member States in taking appropriate actions and monitoring air quality standards that are established in the current Ambient Air Quality Directive 2008/50/EC and fourth daughter Directive 2004/107/EC for a range of pollutants including particulate matter ($PM_{10}$; aerosols with an aerodynamic diameter less than 10 μm), nitrogen dioxide ($NO_2$), ozone ($O_3$), and benzo[a]pyrene. These directives provide the current framework for the control of ambient air pollution in the EU. Effective action to reduce air pollution and its impacts requires a good understanding of sources, atmospheric transport, and transformation mechanisms, and the effects of these atmospheric pollutants on humans and ecosystems [2]. Pending issues are the contributions of biomass emissions to $PM_{10}$ and benzo[a]pyrene in the atmosphere, as well as the formation of secondary aerosols and their influence on PM and their toxicity [3].



Organic aerosol (OA) constitutes a highly variable fraction of PM. Understanding of the different chemical, physical, and toxicological properties of its components is important for applying effective mitigation strategies. OA can be directly emitted by primary sources (Primary Organic Aerosol, POA) or originated by oxidation and condensation of Volatile Organic Compounds (VOCs) leading to Secondary Organic Aerosol (SOA) formation [6]. Primary organic sources in urban areas are related to combustion, cooking, industrial activities, soil and road dust, among others, while biomass burning and soil dust particles are dominant in rural areas [7–9]. The presence of oxidants in the atmosphere such as $O_3$, NOx, and hydroxyl radicals interacts with primary emitted organics in complex reactions to form oxygenated products, such as dicarboxylic acids [10–12]. The oxygenated organic fraction in atmospheric PM can range from 20 to 90%, evidencing the need of understanding the role of the processes involved in SOA formation [13].

The presence of VOCs and oxidants in the atmosphere is related to the occurrence and origin of high $O_3$ concentration. $O_3$ peak concentrations in combination with PM events in summer have been related to the geographical situation in the Western Mediterranean Basin due to its abrupt orography and potential emission sources for VOCs in the urbanized and industrialized coastal areas, and forested areas [14–16]. The synoptic meteorological conditions in the area is commonly characterized by weak horizontal atmosphere pressure (anticyclone) in combination with an active sea–mountain breeze system. This leads to a recirculation of air pollutants and to an accumulation of PM and $O_3$ in the regional atmosphere, especially in mountain valleys [9,17–21].

Molecular organic tracer compound can be used to identify the presence and magnitude of POA and SOA contributions to PM [8,9,22–27]. These compounds can include Polycyclic Aromatic Hydrocarbons (PAHs) and quinones from incomplete combustion of organic matter, hopanes from motorized vehicles emissions, and anhydrosugars, such as levoglucosan, galactosan, and mannosan, from biomass burning. On the other hand, 2-methyltetrols and 2-methylglyceric acid are SOA tracers of isoprene oxidation, while cis-pinonic acid, pinic acid, 3-hydroxyglutaric acid, and 3-methyl-1,2,3-butanetricarboxylic acid (MBTCA) are SOA tracers of $\alpha$-pinene oxidation.

In this study, these molecular organic tracer compounds have been analyzed in $PM_{10}$ samples from three distinctive air quality monitoring stations in 2019. These sites were an urban traffic station (Barcelona), a sub-urban background station (Manlleu), and a rural background station (Bellver de Cerdanya). The selection of the sites was based on the different geographical characteristics and presence of potential source emissions that often leads to exceedance of air quality standards. Five sampling days were selected in these three sites according to $PM_{10}$ filter sample availability and the contrasting air quality, and meteorological conditions. The selected samples allow an assessment of the similarities and differences in OA concentration and source contributions in the three stations by using the data set of the molecular organic tracer compounds. Although these compounds normally only present a small fraction of the OA mass, the relatively small number of tracer compounds can be used in chemometric methods with the aim to reconstruct the emission sources and processes that contribute to OA. There are different source apportionment techniques that can be applied to the data of chemical compounds, such as Chemical Mass Balance (CMB), Principle Component Analysis (PCA), or Positive Matrix Factorization (PMF). CMB uses emission source profiles of the compounds, and is unable to identify unknown sources or processes. PCA uses orthogonal constraints, and the environmental interpretation of the results is not straightforward. On the other hand, PMF uses constraints that are more natural to interpret, such as non-negativity and uncertainty estimations in a non-linear distribution and source profile. In the present study, the Multivariate Curve Resolution—Alternating Least Square (MCR-ALS) method was applied [28–30]. MCR-ALS uses alternating least square optimization under non-negativity constraints that produces better source/process profiles than PCA, and has shown to give results that are equivalent to PMF [31,32]. The MCR-ALS method was successfully used in previous studies with air quality indicators [33], and is a powerful tool for the source apportionment analysis

of the OA [9,22]. Health risk for chronic exposure to these atmospheres was evaluated with the WHO air quality standards for $PM_{10}$, $NO_2$, $O_3$, and benzo[a]pyrene, in relation to annual mean concentrations in the stations. Toxicity tests of the $PM_{10}$ filter extracts were performed on epithelial lung cell line A549. The information obtained in this study can be useful for mitigation strategies to improve the air quality, and also in other areas.

## 2. Materials and Methods

### 2.1. Studied Air Quality Stations and PM10 Sample Selection

The studied sites (Figure 1) are air quality monitoring stations that are included in the Network of Surveillance and Prevention of Air Pollution (XVPCA) of Generalitat de Catalunya in collaboration with the Public Health Agency of Barcelona. Figure 2a–d shows the yearly mean concentrations of $PM_{10}$, $NO_2$, $O_3$, and benzo[a]pyrene in the studied stations since 2011. The sampling station in Barcelona (Eixample; 41.38532° N; 2.15380° E; 26 masl) is a traffic site that is located in a densely populated neighborhood (35.000 inhab./km²) with about 6.000 motorized vehicles/km². This station registers the highest concentrations of air quality indicators for $NO_2$, $PM_{10}$, and benzo[a]pyrene in the city (Figure 2a–d). The sampling station in Manlleu (42.00331° N; 2.28730° E; 460 masl) is a sub-urban background site that is located next to the regional hospital. The town is moderately populated (1.230 inhab./km²) and situated in an extended plain valley with agricultural fields and life stock farms. The plain valley is surrounded by middle-altitude mountains and connected by river valleys (Congost and Ter) to the plains of Barcelona and Vallès in the south, and Girona in the east. This station registers one of the highest concentrations of $O_3$ and benzo[a]pyrene on the Iberian Peninsula and southern Europe (Figure 2a–d). The sampling station of Bellver de Cerdanya (42.36828° N; 1.77680° E; 1060 masl) is a rural background site that is located next to the local primary school. This small town (2.000 inhabitants) is situated in a wide mountain valley in the Pyrenees and is surrounded by mountains that are almost 3000 m high. This station registers high concentrations of $O_3$ and benzo[a]pyrene, but low concentrations of $NO_2$ (Figure 2a–d).

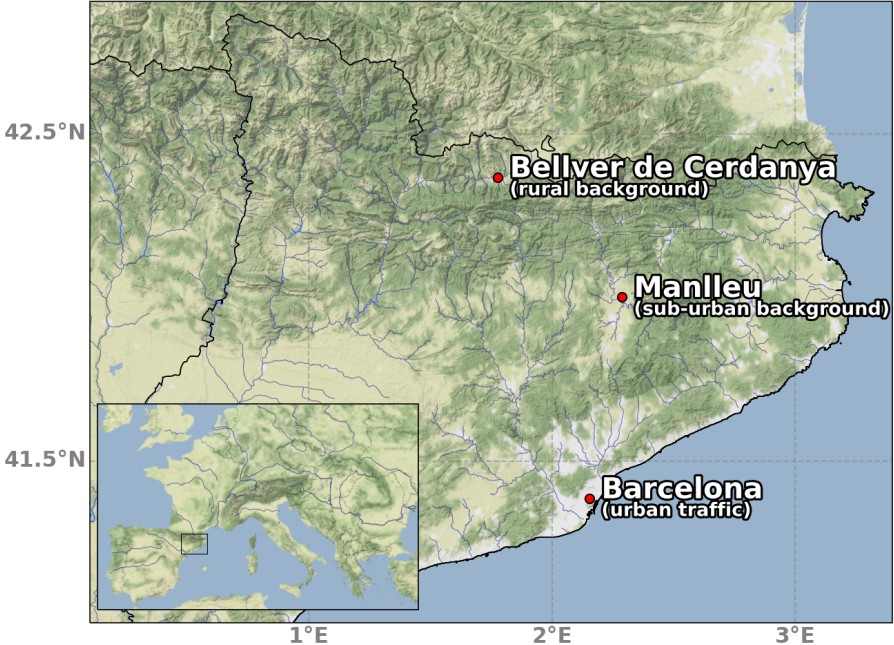

**Figure 1.** Geographical location of the XVPCA stations in Catalonia (Spain).

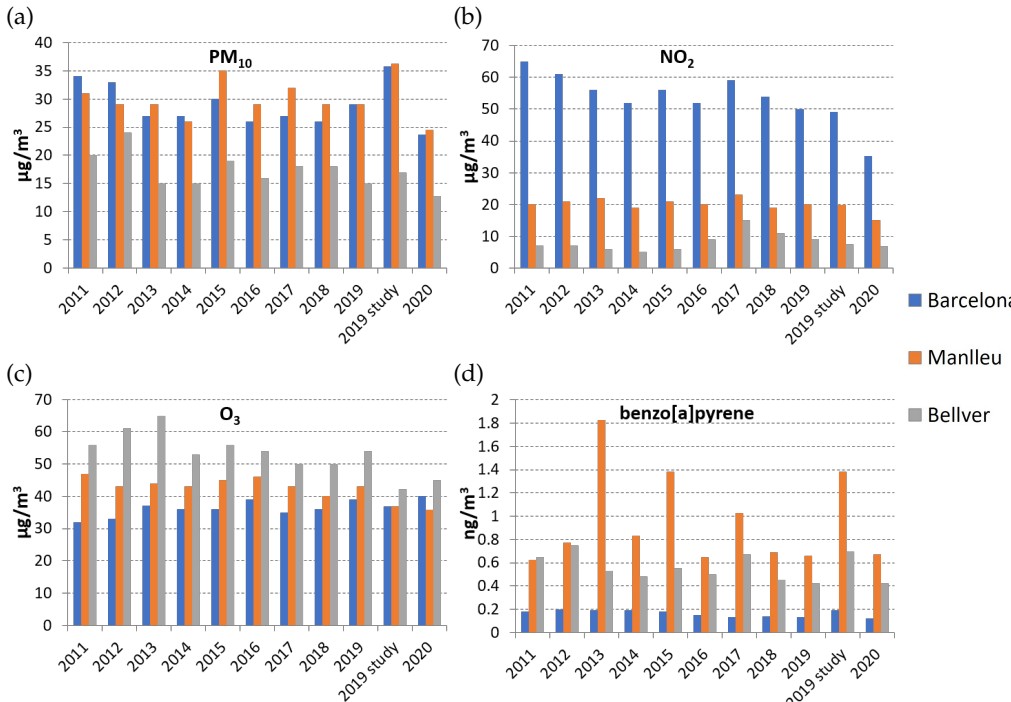

**Figure 2.** Mean concentrations of (**a**) PM$_{10}$, (**b**) NO$_2$, (**c**) O$_3$, and (**d**) benzo[a]pyrene in the air quality monitoring stations of Barcelona (Eixample), Manlleu, and Bellver de Cerdanya run by the regional authorities. The mean concentrations of the fifteen samples that were studied for source apportionment are indicated by "2019 study".

Figure 3a shows the mean 24-h concentrations of the air quality indicators, PM$_{10}$, NO$_2$, and O$_3$, for the period from August to December 2019 in the three studied stations. These mean concentrations were based on hourly data. There were only a few hours that data were missing, and these missing data points were assumed to be the average of the last and next known data point. This simple method provides good results in data sets with few missing random data points [34]. In the present study, these data were used to select the days from which collected PM$_{10}$ filters were to be analyzed on organic molecular tracer compounds and PM toxicity (Figure 3b). To select these days, Multivariate Curve Resolution—Alternating Least Square (MCR-ALS) analysis was applied on the augmented data matrix of the PM$_{10}$, NO$_2$, and O$_3$ concentrations in the three stations. This method decomposes the data matrix into the product of two matrices containing the information of the resolved components that is similar to Principle Component Analysis (PCA). In contrast to PCA, the factors found with MCR-ALS are not orthogonal and the imposition of constraints, such as non-negativity via alternating least squares algorithm, gives a solution with a physically meaningful interpretation [28–30]. MCR-ALS is based on a bilinear decomposition of the original data set. In matrix form, it is expressed as D(I × J) = U(I × N)V$^T$ (N × J ) + E(I × J), where D is the original data array, with I rows (days) and J columns (air quality indicators); U is the matrix of scores of dimensions I × N, where N is the reduced number of components; V$^T$ is the matrix of loadings with dimensions N × J; and E is the matrix of residuals not modeled by the N components. The augmented data matrix was imported into MATLAB for subsequent calculations [28]. The MCR-ALS analysis resolved two components that explained 97% of the variance in the data. The first component had higher loadings of PM$_{10}$ and NO$_2$, while the second component had high O$_3$ loadings (Figure A1). For the selection of the days from these results, the following criteria were applied: (1) High PM$_{10}$ and NO$_2$ concentrations in the three stations, (2) high O$_3$ concentrations in the three stations, and (3) low PM$_{10}$ in the three stations. To select the sample days for criteria 1 and 2 from the two resolved components, the cut-off for the score values was set at 5.6 (Figure A2). The anti-correlation between

the component score values was used to distinguish the third criteria, with overall low $PM_{10}$, between these cut-off score values (Figure A2). This resulted in 55 days that could be divided over the three groups. However, the days that $PM_{10}$ filter samples were collected simultaneously in all the three stations were very limited. This was the case in five of these days; 2nd of August (02/08; group 2); 16th of November (16/11; group 3); 20th of November (20/11; group 1); 18th of December (18/12; group 1); 22nd of December (22/12; group 2) (Figure 3b).

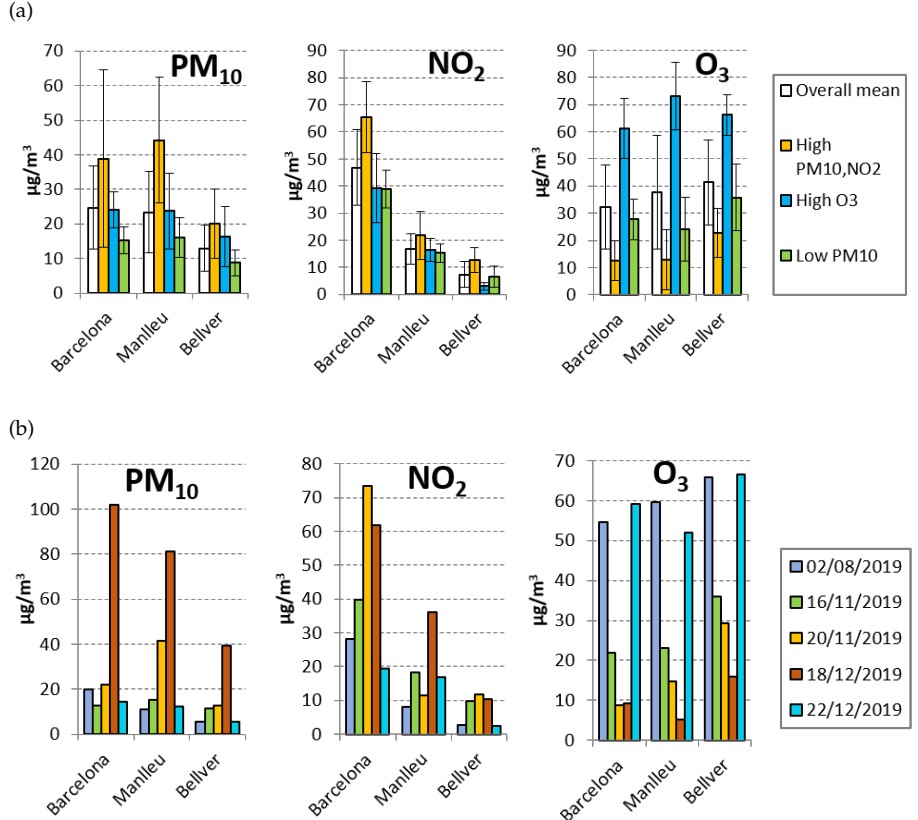

**Figure 3.** (**a**) Mean 24-h concentrations (and standard deviation) of $PM_{10}$, $NO_2$, and $O_3$ of the samples that were resolved in the MCR-ALS analysis into 3 groups of different criteria: (1) high $PM_{10}$, $NO_2$ concentrations in the tree stations; (2) high $O_3$ concentrations in the three stations; (3) low $PM_{10}$ in the three stations. (**b**) Mean 24-h concentrations of $PM_{10}$, $NO_2$, and $O_3$ of the 5 days that were selected for organic tracer analysis and toxicity tests of the $PM_{10}$ filter sample.

### 2.2. Air Mass Back-Trajectories

The air mass back-trajectories of the five days were calculated at 100 m, 500 m, and 1000 m a.g.l. for the three stations with the NOAA HYSPLIT model [35,36]. The meteorological model fields that were used to compute the trajectories were simulated with the fully compressible non-hydrostatic Weather Research and Forecasting model (WRF-ARW) version 3.5 [37] coupled in the ARAMIS air quality modeling system [21] with a resolution of 3 km × 3 km. Vertical coordinates, grid type, configuration domains, boundary conditions, and model physics configuration are described in more detail elsewhere [21,38]. The WRF-ARW model also provides data of the planetary boundary layer height (PBLH), which helps to characterize the air pollution dispersion conditions. Furthermore, to consider larger-scale advections, 72 h backward trajectories were computed at 100 m and 500 m with the online version of HYSPLIT with the gridded meteorological data from NCEP Global Forecast System (GFS) archive with a resolution of 0.25° × 0.25°. In addition, synoptic meteorological patterns of the selected days were analyzed with the Climate Forecast System Reanalysis (CFSR) model with a resolution of 0.5° [39].

### 2.3. Organic Molecular Tracer Analysis

Organic molecular tracer compounds were analyzed in the fifteen selected $PM_{10}$ filters (five from each station). These compounds consisted of levoglucosan, mannosan, and galactosan as indicators of biomass burning emissions; dicarboxylic acids and polyols of SOA formation (including SOA products from isoprene and α-pinene oxidation), and hopanes of traffic emissions. Moreover, thirteen PAH compounds, including benzo[a]pyrene, as well as three quinones (oxygenated PAHs), were analyzed in these samples to calculate toxic equivalent quotients of the PAHs and to determine the relative composition of the isomers that can be related to source emissions [23,40,41]. One quarter of the $PM_{10}$ filters were spiked with deuterated PAHs, levoglucosan-D7 and succinic acid-D4, and extracted in a solvent mixture of methanol : dichloromethane ($3 \times 10$ mL (1:1 $v/v$)) by ultra-sonication. Extracts were filtered and concentrated by roto-evaporation and with a gentle stream of nitrogen to a final volume of 0.5 mL. An aliquot of 25 μL was evaporated to dryness and treated with 25 μL BSTFA + 10 μL pyridine to obtain the trimethylsilyl-derivatives (TMS) of the polar compounds (saccharides and acids), prior to their analysis by gas chromatography coupled to mass spectrometry (GC-MS). The remaining extract was extracted with $2 \times 0.5$ mL of n-hexane and concentrated under a gentle stream of nitrogen to 25 μL for the analysis of PAHs and hopanes by GC-MS. This method is described in more detail elsewhere [9,42].

### 2.4. Source Apportionment of Organic Aerosol

In the present study, the normalized data matrix containing organic tracer compounds in the different samples was subjected to MCR-ALS under non-negativity constraint, resulting in the resolution of several components, each one represented by two profiles that described, on one hand, their contribution to the different samples (score values), and on the other hand, their organic tracer compounds profiles (loadings) [28–30]. MCR-ALS has previously been applied successfully for the source apportionment of organic aerosol in urban and rural atmospheres [9,22].

### 2.5. PM Toxicity

In order to study PM toxicity, the epithelial lung cell line A549 was obtained from the American Type Culture Collection (ATCC CCL-185) and cultured using Dulbecco's Modified Eagle's Medium (DMEM) with Ultraglutamine (BE12-604F/U1, Lonza, Basel, Switzerland) supplemented with 10% of fetal bovine serum (10270-106, Gibco, Thermo Fisher, Waltham, United States). Cells were cultured in 75 cm$^2$ polystyrene vented flasks, and incubated at 37 °C in a humidified incubator set at 5% $CO_2$, and passaged every 3–4 days.

The filter extracts that were used to expose A549 cultures, were obtained from a 1/8 fraction of each $PM_{10}$ filter (corresponding to 90 m$^3$ of filtered air). In addition to the 15 filter samples, 1 blank filter was used and subjected to the same extraction procedure. Each piece of filter was cut in small pieces and submerged in 4 mL of supplemented DMEM (the same used for cell culture) in a 15 mL plastic tube. Tubes were vortexed for 1 min and then sonicated in a bath for 15 min. The extraction media were then transferred to clean tubes, and 2 mL of supplemented DMEM were added to the filter tube to repeat the procedure and increase the recovery of pollutants from the filter. The extracts were sonicated together for 15 additional minutes to disaggregate the potential particles released from the filter. Tubes were centrifuged at 15,000 rpm for 10 min, and supernatants were recovered in new tubes. These supernatants were sterilized under UV-radiation for 1 h prior to the cell culture exposure.

To prepare the viability assays, A549 cells were seeded in flat-bottomed 96-well plates (Nunc) at 0.1 million cells/mL. After 24 h, cells were attached to the plates and they were ready to be treated with the 15 samples and blank extracts. Each well was filled with 100 μL of media. The high exposure dose consisted in 100 μL of the previously obtained extraction media (this dose was considered 15 m$^3$ eq. air/mL). The other six decreasing concentrations

were prepared by sequential 1:2 dilutions. Control cells were incubated with supplemented DMEM without any sample extract. Each condition was assayed in triplicate. Culture plates were then incubated for 72 h at standard conditions. After this time, the resazurin cell viability assay was performed (CellTiter-Blue, Promega) following the manufacturer indications. After 4 h of additional incubation in the presence of resazurin reagent, cell plates were read in a fluorescence microplate reader (Infinite M Plex, Tecan) at 560/590 nm excitation/emission wavelengths. IC50 calculations and curves were performed using GraphPad Prism 8 using the non-linear regression variable slope model.

## 3. Results and Discussion

### 3.1. Atmospheric Conditions

The synoptic conditions on 02/08 were characterized by anticyclonic conditions with small pressure gradients, which favored breeze circulations in the area. On 16/11, a 500 hPa trough in the Iberian Peninsula gave a situation of atmospheric instability. On 20/11, small pressure gradients at surface and 500 hPa resulted in stagnant air conditions. On 18/12, high surface pressures on the Iberian Peninsula also created stagnant conditions in the studied area and an African dust outbreak at high levels occurred due to a low pressure in the north of Africa. On 22/12, a strong pressure gradients gave strong western winds in all southern Europe. In this situation, strong Föhn wind affected the studied area, favoring the mixing of air from the free troposphere. All these synoptic conditions are in agreement with the long-scale and short-scale trajectories of the sample days as well as the meteorological conditions in the stations (Table 1).

The lowest wind speeds were on 02/08, 18/12, and 20/11, the days with stagnant meteorological conditions, as previously described. The strongest winds were on 22/12, when some hourly-mean winds exceeded 10 m/s. For this last day, standard deviation in wind direction and wind speed was relatively small, indicating strong westerlies during all day (Table 1).

**Table 1.** Long scale advections of the studied days and daily average of modeled 10 m Wind Speed (WS) and Wind Direction (WD) with the Standard Deviation (sd).

| Day | Long-Scale Advection | Barcelona | | Manlleu | | Bellver | |
|---|---|---|---|---|---|---|---|
| | | WD (sd) (Degrees) | WS (sd) (m/s) | WD (sd) (Degrees) | WS (sd) (m/s) | WD (sd) (Degrees) | WS (sd) (m/s) |
| 2/8 | Weak Atlantic— Gulf of Lyon | 244 (79) | 2 (1) | 61 (60) | 2 (1) | 278 (47) | 3 (1) |
| 16/11 | Atlantic | 314 (46) | 6 (3) | 214 (37) | 3 (2) | 254 (39) | 3 (2) |
| 20/11 | Continental | 316 (37) | 2 (1) | 202 (8) | 5 (2) | 242 (21) | 3 (2) |
| 18/12 | African | 336 (27) | 3 (1) | 178 (88) | 1 (1) | 311 (61) | 2 (1) |
| 22/12 | Strong Atlantic | 269 (11) | 9 (2) | 240 (11) | 8 (2) | 269 (14) | 11 (3) |

PBLH data obtained from WRF-ARW model confirmed the atmospheric conditions expressed by the synoptic meteorological patterns. Figure 4 shows the PBLH profiles for the Manlleu station, which is representative of the studied region. The highest PBLH were reached in summer (02/08) when radiation and convective activity are stronger followed by 22/12 when the strong winds favored turbulence and the growing of the mixing layer. For this last day, PBLH was also high at night-time due to the strong wind. Lower mixing layer heights were modeled for 16/11 when Atlantic advection was weaker. On the other hand, the lowest mixing layer were observed on 20/11 and 18/12 when stagnant atmospheric conditions were dominant.

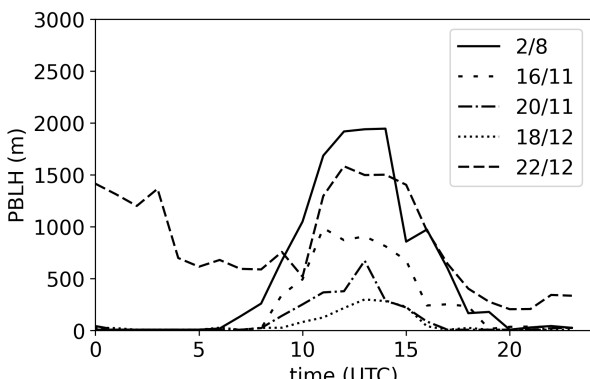

**Figure 4.** Modeled PBLH profiles in Manlleu for all the studied days.

### 3.2. Air Quality Indicators and Organic Molecular Tracer Concentrations

The annual mean concentrations of the air quality indicators (Figure 2a–d) in the three sites fluctuate among years but there is no clear up-going or down-going trend. Nevertheless, in 2020 the $PM_{10}$ and $NO_2$ mean concentrations were lower in the urban and sub-urban station due to the COVID-19-related lockdown (March to May) and consequential traffic reduction [43] while benzo[a]pyrene did not present significant changes. On the other hand, $O_3$ mean concentrations decreased in the rural and sub-urban stations and increased in the urban one, probably as a result of the distinct ozone precursors regimes [44,45]. Despite small inter-annual changes over the past decade, important concentration fluctuations occurred among the selected days in this study (Figure 3b). In fact, the selection of these days was based on these (intra-annual) variations, in order to obtain an insight on the composition of organic molecular tracer compounds and the toxicity of $PM_{10}$ in these stations. Table 2 shows the concentration of the air quality indicators and the organic molecular tracer compounds in the $PM_{10}$ samples, while Figure 5 shows representative air-mass back-trajectories for each station and sample day. The trajectories were associated with the hourly $PM_{10}$ concentrations that were registered in the station (Figure 5).

On day 02/08, the overall 24-h mean $O_3$ concentrations were high (>60 µg/m$^3$). The acids and polyols that are related to biogenic SOA formation from isoprene and α-pinene oxidation (3HGA, MBTCA, 2MGA, 2MT1, 2MT2) were ten times higher on this day compared to the other days, in all the three stations. The highest biogenic SOA concentrations were observed in the rural background station Bellver de Cerdanya (Pyrenees), coinciding with the presence of large extensions of forests and vegetation in the area. This summer day showed overall low anhydro-saccharide concentrations (biomass burning tracers), as well as the lowest hopane (traffic tracer) concentrations. Nevertheless, the hopane concentrations were always higher in the urban traffic station of Barcelona than those observed in any of the studied days in Bellver de Cerdanya. This trend was also observed in the $NO_2$ concentration, which is also an indicator for traffic emissions. In fact, a Pearson-correlation test of the hopane concentrations with the $NO_2$ concentrations showed very strong correlations ($R^2 = 0.8$; $p < 0.01$). For 02/08, a coastal advection is observed in almost all trajectories (Figure 5a), which is usual in summer. These local breeze conditions contribute to background ozone and circulation of pollutants, which might enhance the aging of organic pollutants [46]. Moreover, many trajectories in Bellver and Manlleu passed through Barcelona in the previous hours, via the Congost river valley (in case of Manlleu) and the Llobregat river valley (in case of Bellver). The trajectories were following the inland breeze circulations, which may transport ozone precursors to the inland stations.

**Table 2.** Concentration of $PM_{10}$, $NO_2$, and $O_3$ in $\mu g/m^3$ and the analyzed organic compounds for all samples in $ng/m^3$.

| | BCN | | | | | Manlleu | | | | | Bellver | | | | |
|---|---|---|---|---|---|---|---|---|---|---|---|---|---|---|---|
| | 02/08/2019 | 16/11/2019 | 20/11/2019 | 18/12/2019 | 22/12/2019 | 02/08/2019 | 16/11/2019 | 20/11/2019 | 18/12/2019 | 22/12/2019 | 02/08/2019 | 16/11/2019 | 20/11/2019 | 18/12/2019 | 22/12/2019 |
| $PM_{10}$ | 28 | 13 | 22 | 102 | 14 | 31 | 15 | 42 | 81 | 12 | 16 | 11 | 13 | 39 | 6 |
| $NO_2$ | 51 | 40 | 74 | 62 | 19 | 16 | 18 | 12 | 36 | 17 | 3 | 10 | 12 | 10 | 2 |
| $O_3$ | 85 | 22 | 9 | 9 | 59 | 90 | 23 | 15 | 5 | 52 | 63 | 36 | 29 | 16 | 67 |
| succinic_acid (SA; $m/z$ 247) | 5.15 | 3.07 | 5.27 | 9.98 | 2.72 | 6.72 | 2.80 | 6.81 | 9.62 | 2.65 | 15.48 | 12.83 | 13.48 | 15.80 | 16.41 |
| glutaric_acid (GLU; $m/z$ 261) | 1.44 | 0.51 | 0.93 | 4.31 | 1.50 | 1.17 | 0.56 | 2.23 | 3.71 | 0.52 | 1.89 | 3.08 | 3.33 | 5.50 | 2.77 |
| azealic_acid (AZA; $m/z$ 317) | 6.36 | 3.92 | 14.06 | 8.90 | 3.27 | 6.14 | 4.88 | 13.75 | 15.46 | 2.00 | 6.69 | 11.13 | 12.20 | 24.39 | 6.25 |
| phthalic acid (PHA; $m/z$ 295) | 1.68 | 1.20 | 3.63 | 9.47 | 15.88 | 3.39 | 2.22 | 10.68 | 10.67 | 0.51 | 2.48 | 4.33 | 4.41 | 8.02 | 2.23 |
| cis pinonic_acid (CPA; $m/z$ 171) | 0.53 | 1.27 | 1.84 | 3.22 | 2.20 | 0.61 | 1.51 | 1.21 | 0.57 | 0.47 | 0.84 | 2.81 | 2.70 | 3.85 | 2.53 |
| malic acid (MA; $m/z$ 233) | 8.19 | 0.90 | 2.80 | 1.69 | 1.90 | 39.11 | 5.09 | 6.86 | 3.11 | 0.63 | 47.06 | 2.58 | 2.62 | 2.28 | 2.11 |
| 3-hydroxyglutaric (HGA; $m/z$ 249) | 5.05 | 1.49 | 2.13 | 0.44 | 0.69 | 11.82 | 0.85 | 3.27 | 2.10 | 0.29 | 24.58 | 1.31 | 1.56 | 1.04 | 0.68 |
| 3-methyl-1,2,3-butanetricarboxylic acid (MBTCA: $m/z$ 405) | 5.11 | 0.63 | 0.47 | 0.11 | 1.29 | 13.51 | 1.00 | 1.74 | 0.56 | 0.07 | 13.63 | 0.15 | 0.46 | 0.25 | 0.16 |
| 2-metylglyceric_acid (MGA; $m/z$ 219) | 24.45 | 0.30 | 0.72 | 2.18 | 0.62 | 26.05 | 0.92 | 1.58 | 1.04 | 0.27 | 51.62 | 1.24 | 1.76 | 1.97 | 1.14 |
| 2-methylthreitol (2MT1; $m/z$ 219) | 3.75 | 0.28 | 1.04 | 1.79 | 0.21 | 2.77 | 0.70 | 2.01 | 0.61 | 0.08 | 20.27 | 2.90 | 2.19 | 7.97 | 1.10 |
| 2-methylerythritol (2MT2; $m/z$ 219) | 12.44 | 0.53 | 2.29 | 6.50 | 0.61 | 15.72 | 2.17 | 3.16 | 2.04 | 0.12 | 101 | 4.40 | 3.07 | 7.60 | 2.15 |
| galactosan (GAL; $m/z$ 217) | 0.17 | 6.41 | 5.76 | 10.89 | 1.29 | 0.32 | 29.28 | 92.9 | 161 | 1.62 | 1.06 | 46.26 | 49.00 | 78.9 | 6.00 |
| mannosan (MAN; $m/z$ 204) | 0.37 | 7.75 | 8.15 | 12.12 | 1.18 | 0.58 | 29.03 | 99.3 | 343 | 6.91 | 7.10 | 85.3 | 97.7 | 117 | 17.99 |
| levoglucosan (LEV; $m/z$ 204) | 5.52 | 74.4 | 91.8 | 135 | 17.14 | 5.90 | 400 | 1384 | 4978 | 96.9 | 36.90 | 829 | 963 | 1168 | 290 |
| 17a(H)21$\beta$(H)-29-norhopane (norHOP; $m/z$ 191) | 0.091 | 0.173 | 0.527 | 0.460 | 0.137 | 0.021 | 0.028 | 0.175 | 0.181 | 0.009 | 0.008 | 0.025 | 0.048 | 0.065 | 0.011 |
| 17a(H)21$\beta$(H)-hopane (HOP; $m/z$ 191) | 0.232 | 0.230 | 0.533 | 0.556 | 0.124 | 0.038 | 0.030 | 0.179 | 0.150 | 0.010 | 0.012 | 0.031 | 0.051 | 0.056 | 10.012 |
| benzo[a]fluorenone (BAF; $m/z$ 230) | 0.028 | 0.045 | 0.065 | 0.069 | 0.023 | 0.008 | 0.041 | 0.164 | 0.377 | 0.014 | 0.004 | 0.075 | 0.075 | 0.114 | 0.013 |
| benzo[b]fluorenone (BBF; $m/z$ 230) | 0.011 | 0.048 | 0.066 | 0.060 | 0.015 | 0.005 | 0.064 | 0.316 | 0.765 | 0.017 | 0.009 | 0.174 | 0.147 | 0.290 | 0.024 |
| benzanthrone (BA; $m/z$ 230) | 0.014 | 0.051 | 0.087 | 0.077 | 0.018 | 0.008 | 0.155 | 0.712 | 1.536 | 0.034 | 0.005 | 0.405 | 0.336 | 0.487 | 0.056 |
| benz[a]anthracene (BAA; $m/z$ 228) | 0.04 | 0.07 | 0.25 | 0.22 | 0.05 | 0.02 | 0.16 | 1.46 | 1.48 | 0.04 | 0.01 | 0.39 | 0.38 | 0.59 | 0.05 |
| chrysene + triphenylene (C + T; $m/z$ 228) | 0.06 | 0.15 | 0.36 | 0.41 | 0.08 | 0.06 | 0.28 | 1.94 | 3.00 | 0.07 | 0.02 | 0.59 | 0.49 | 1.01 | 0.09 |
| benzo[b + j]fluoranthene (BBJFL; $m/z$ 252) | 0.08 | 0.22 | 0.67 | 0.60 | 0.05 | 0.05 | 0.61 | 5.94 | 6.37 | 0.06 | 0.02 | 1.87 | 1.47 | 3.69 | 0.14 |
| benzo[k]fluoranthene (BKFL; $m/z$ 252) | 0.03 | 0.09 | 0.24 | 0.18 | 0.05 | 0.02 | 0.18 | 1.84 | 1.83 | 0.03 | 0.01 | 0.58 | 0.49 | 1.17 | 0.05 |
| benzo[e]pyrene (BEP; $m/z$ 252) | 0.14 | 0.19 | 0.53 | 0.40 | 0.11 | 0.06 | 0.34 | 3.01 | 3.53 | 0.05 | 0.03 | 0.95 | 0.77 | 1.42 | 0.13 |
| benzo[a]pyrene (BAP; $m/z$ 252) | 0.06 | 0.11 | 0.40 | 0.33 | 0.04 | 0.04 | 0.25 | 3.35 | 3.24 | 0.03 | 0.01 | 0.97 | 0.82 | 1.60 | 0.08 |
| indeno[123cd]pyrene (IP; $m/z$ 276) | 0.05 | 0.15 | 0.35 | 0.31 | 0.05 | 0.04 | 0.41 | 2.33 | 2.49 | 0.07 | 0.02 | 1.00 | 0.65 | 1.13 | 0.14 |
| dibenzo[ah]anthracene (DAA; $m/z$ 278) | 0.01 | 0.03 | 0.03 | 0.03 | 0.02 | 0.01 | 0.09 | 0.62 | 0.62 | 0.02 | 0.01 | 0.13 | 0.13 | 0.27 | 0.03 |
| benzo[ghi]perylene (BGP; $m/z$ 276) | 0.11 | 0.27 | 0.65 | 0.73 | 0.10 | 0.05 | 0.52 | 3.86 | 4.09 | 0.09 | 0.03 | 1.16 | 0.92 | 1.60 | 0.18 |
| ΣPAH | 0.6 | 1.3 | 3.5 | 3.2 | 0.6 | 0.3 | 2.8 | 24.4 | 26.7 | 0.5 | 0.2 | 7.7 | 6.1 | 12.5 | 0.9 |
| BAP/(BAP + BEP) | 0.29 | 0.38 | 0.43 | 0.45 | 0.25 | 0.42 | 0.43 | 0.53 | 0.48 | 0.42 | 0.34 | 0.50 | 0.51 | 0.53 | 0.38 |
| IP/(IP + BGP) | 0.31 | 0.35 | 0.35 | 0.30 | 0.34 | 0.47 | 0.44 | 0.38 | 0.38 | 0.42 | 0.45 | 0.46 | 0.42 | 0.41 | 0.44 |
| TEQ_PAH | 0.2 | 0.3 | 0.8 | 0.7 | 0.1 | 0.1 | 0.9 | 8.0 | 8.0 | 0.1 | 0.0 | 2.1 | 1.9 | 3.8 | 0.3 |

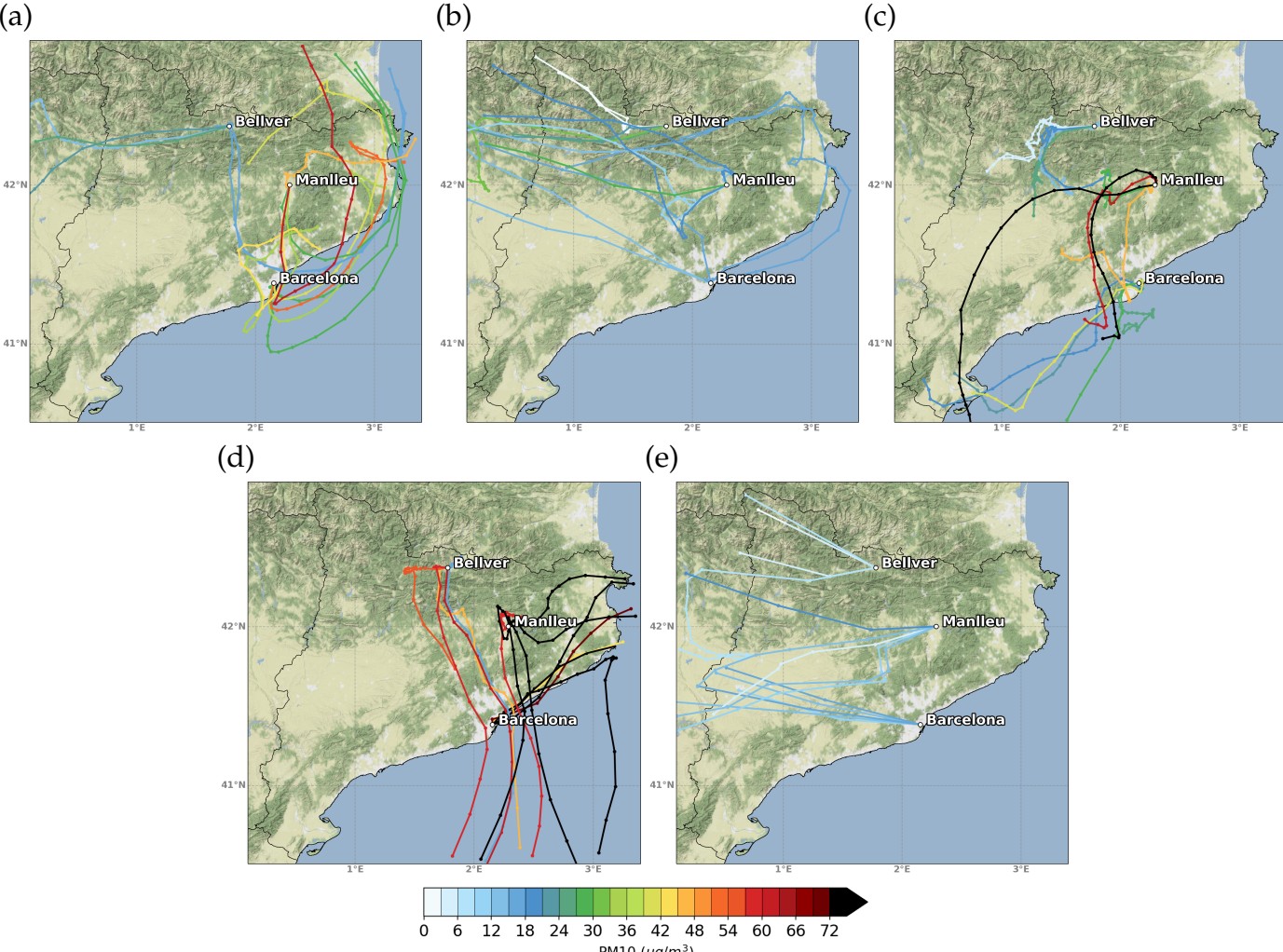

**Figure 5.** Representative hourly air-mass backward trajectories at 100 m related with $PM_{10}$ concentrations automatically measured at the station the hour of arrival. (**a**) 02/08/2019, (**b**) 16/11/2019, (**c**) 20/11/2019, (**d**) 18/12/2019, and (**e**) 22/12/2019.

On day 16/11, the overall $PM_{10}$ concentrations were low (<15 μg/m$^3$). The hopane traffic tracer concentrations were similarly low as in the former summer day. Notwithstanding, the urban traffic station in Barcelona showed almost ten times higher levels than in the sub-urban station of Manlleu and the rural background station of Bellver de Cerdanya. On the other hand, these two background stations showed about ten times higher biomass burning tracer concentrations than the urban traffic site. The biogenic SOA tracer compounds (acids and polyols) were about ten times lower than in summer. Regarding the associated air-mass trajectories (Figure 5b), a clear Atlantic air-mass origin was observed with some coastline contributions for the urban station of Barcelona (but also with Atlantic origin). These kind of Atlantic advections are usually related with good air quality conditions [47,48] since cleaner air arrives from non-industrialized areas, although local emissions influence the stations.

On day 20/11, the $PM_{10}$ concentrations were high in the urban traffic (Barcelona; 22 μg/m$^3$) and sub-urban (Manlleu; 42 μg/m$^3$) stations, but low in the rural background station (13 μg/m$^3$). Traffic related tracers, $NO_2$ and hopanes, were also high in Barcelona, but low $NO_2$ concentration was found in Manlleu. On the other hand, the biomass tracer concentration was high in Manlleu (levoglucosan = 1400 ng/m$^3$). The rural background

station in Bellver de Cerdanya was also dominated by biomass burning, but the $PM_{10}$ concentrations were low. For the particular case of Bellver de Cerdanya, low $PM_{10}$ were probably related to local circulation of air masses in the nearby mountain valleys, as observed in the 24 h trajectories (Figure 5c). Conversely, most of the trajectories for Barcelona came from the south coast of the territory. In Manlleu, trajectories were divided by those recirculating in the Llobregat valley and those coming from the south-west of Catalonia. These source areas host several industries that can impact the air quality [49] and may influence the stations of Barcelona and Manlleu in combination with local emissions.

On day 18/12, the $PM_{10}$ concentrations were high in the three stations (five times higher than 20/11), and $NO_2$ was high in the urban traffic Barcelona and the sub-urban Manlleu stations (similar to 20/11), but $O_3$ was low in the three sites (<16 $\mu g/m^3$). On these days, the biomass burning tracer concentrations were the highest observed in Manlleu (levoglucosan = 5000 $ng/m^3$), in Bellver (levoglucosan = 1200 $ng/m^3$), and in Barcelona (levoglucosan = 135 $ng/m^3$). The hopane traffic tracer concentrations in the stations were similar to those of day 20/11. The high $PM_{10}$ concentrations observed on this day were related to Saharan dust intrusion. This was confirmed by HYSPLIT trajectories that showed an origin of the air masses in the north of Africa. This dust was transported from the south towards Bellver de Cerdanya and Manlleu, or following the coastline for Barcelona (Figure 5d). Furthermore, a low PBLH was observed on this day, and this restricted vertical dispersion resulted in high $PM_{10}$ concentrations throughout the day.

On day 22/12, the $PM_{10}$ and $NO_2$ concentrations were low (<16 $\mu g/m^3$ and <19 $\mu g/m^3$, respectively), and high $O_3$ concentrations were high in the three stations (>50 $\mu g/m^3$). The concentrations of the organic tracer compounds were low in all stations, although the biomass burning tracer concentrations were still dominating the aerosol composition in Manlleu and Bellver de Cerdanya. This day was characterized by North Atlantic advection with strong Föhn winds, that resulted in a mixing of local air and air from the free troposphere that contained low particle concentrations but high $O_3$ levels [50–52]. The trajectory pattern was very similar to that of 16/11 but related with stronger winds since air-mass was transported from the east coast of Canada in 72 h. These winds contributed to vertical mixing resulting in a high PBLH (for winter conditions) in all stations.

These results show that the urban traffic station is dominated by traffic emissions, as expected and in agreement with former studies [22,24]. In addition, all stations have a high contribution of biogenic SOA and $O_3$ in summer as was previously described for a rural background site in the region [53]. The sub-urban and rural stations have a high contribution of biomass burning emissions in colder periods that could be related to elevated $PM_{10}$ concentrations and benzo[a]pyrene concentrations, which is in agreement with results from former studies in similar sites [9,54].

### 3.3. Polycyclic Aromatic Hydrocarbons (PAHs) and Quinones (Oxygenated PAHs)

Traffic emissions and biomass burning emissions from domestic heating and vegetative waste removal in agriculture are potential sources for atmospheric PAHs and quinones [9,40,55]. In order to study the influence of these potential emission sources, the concentrations and relative composition of the individual PAHs were determined in the $PM_{10}$ filter samples (Table 2). On days 16/11, 20/11, and 18/12, the benzo[a]pyrene (BAP) concentrations were also analyzed by the regional authorities and a one-to-one comparison between the results showed strong correlations ($R^2 = 0.98$; $p < 0.01$) with acceptable errors (RSD% < 17%). In the present study, four to six-fused ring PAHs, among them BAP, were detected in all the analyzed samples, and the Toxic Equivalent Quotients (TEQ-PAH) were calculated by using the Toxic Equivalent Factors from Nisbet and LaGoy [41]. Oxygenated PAHs, i.e., benzo[a]fluorenone, benzo[b]fluorenone, and benzathrone, were also detected in all samples. Benzathrone was the dominating compound, followed by benzo[a]fluorenone. The concentrations of the individual PAH compounds were highly correlated ($R^2 > 0.96$; $p < 0.01$) despite important concentration variations among sample days and stations.

In the urban traffic station of Barcelona, the ΣPAH ranged between 0.6 ng/m$^3$ and 3.5 ng/m$^3$; in the sub-urban background station of Manlleu between 0.3 ng/m$^3$ and 26.7 ng/m$^3$; and in the rural background station of Bellver between 0.5 ng/m$^3$ and 12.5 ng/m$^3$. The lowest ΣPAH and TEQ-PAH concentrations were observed in the summer sample of 02/08 and the winter sample of 22/12, both characterized by high O$_3$ concentrations. Nevertheless, the summer sample was influenced by weak North Atlantic advective conditions, and recirculation of regional air pollution in the absence of biomass burning (low biomass burning tracer concentrations). The winter sample was characterized by strong North Atlantic advective conditions and mixing of local air with clean air from the free troposphere. On the other hand, the sampling days with high PM$_{10}$ concentrations in Barcelona and Manlleu (20/11 & 18/12), were also the days with the highest ΣPAH concentrations in both stations (3.5 ng/m$^3$ in Barcelona and 26.7 ng/m$^3$ in Manlleu). TEQ-PAH and quinones concentrations were also high on these days in the two stations, but ten times higher in Manlleu than Barcelona. The high levels of PAH, TEQ-BAP, and quinones could be consequence of the prevailing stagnant weather conditions with low PBLH during these days, as observed in the past [56], and this is also the reason for the elevated PM$_{10}$, PAH, and quinones concentrations in Bellver on sampling day 18/12. Notwithstanding, the urban traffic station was dominated by traffic emissions from fossil fuel combustion (high NO$_2$ and hopane concentrations), while the other stations were dominated by biomass burning emissions (high biomass burning tracer concentrations).

In the past, the ratios of PAH isomers have been used to distinguish between sources, despite inter-source emission rate variations, gas–particle partitioning, and photo-chemically degradation of PAHs in the ambient air [40,57]. Particle bounded PAHs are less susceptible to post-emission influences [57], so the isomeric PAH ratios of BAP/(BAP + BEP) and IP/(IP + BGP) may offer an insight to source emissions. These ratios are higher in biomass burning emissions than in vehicle fossil fuel emissions, due to the different combustion conditions (e.g., temperature, humidity, and availability of oxygen) as well as type of fuel (e.g., gasoline, diesel, and type of wood) [8,58]. The ratios in the urban traffic station in Barcelona were 0.36 ± 0.09 for BAP/(BAP + BEP), while they were 0.45 ± 0.05 and 0.45 ± 0.09 in the sub-urban background station in Manlleu and the rural background station in Bellver, respectively. Ratios of IP/(IP + BGP) were 0.33 ± 0.02 in the traffic station, and 0.42 ± 0.04 and 0.44 ± 0.02 in the sub-urban and rural background stations. These findings indicate that traffic emissions have larger influence on the PM$_{10}$ bounded PAHs in Barcelona, while biomass burning emissions dominate in Manlleu and Bellver. Nevertheless, the isomeric PAH ratios can not be used as stand-alone source indicators for atmospheric PAHs. The apportionment of PAH should be combined at least with a proper source apportionment analysis, such as the Multivariate Curve Resolution—Alternating Least Square (MCR-ALS) method or the Positive Matrix Factorization (PMF) method. Moreover, such analysis includes the possibility to quantify the contributions of sources and SOA processes to the organic aerosol.

### 3.4. Source Apportionment of Organic Aerosol and PAHs

For the source apportionment of the organic aerosol (OA), as well as the contributions of the emission sources to particle bounded PAHs, the MCR-ALS method was applied on the augmented database that contained 25 organic tracer compounds and 15 samples. Again, MCR-ALS is based on a bilinear decomposition of the original data set. In matrix form, it is expressed as D(I × J) = U(I × N)V$^T$ (N × J) + E(I × J), where D is the original data array, with I rows (samples) and J columns (compounds); U is the matrix of scores of dimensions I × N, where N is the reduced number of components; V$^T$ is the matrix of loadings with dimensions N × J ; and E is the matrix of residuals not modeled by the N components. The MCR-ALS method decomposes the data matrix using an alternating least squares algorithm under non-negativity constraint [28–30].

Four clear components were obtained that explained 95% of the variation in the database (Figure 6a–d). The first component was the Traffic OA (Figure 6a), and it contained

91% of the detected hopanes in the samples, linking this component to traffic emissions. 20% of the detected PAHs were presented in this component, showing that traffic emission is not a dominating source for PAHs. Furthermore, this urban component contained 28% of the dicarboxylic acids (SA, GLA, and AZA) and phthalic acid (PHA). In the past, these acids have been observed in urban traffic sites of densely populated areas and related to SOA from traffic emissions, but also other activities, such as food cooking [22]. The second component was the biomass burning OA (BBOA; Figure 6b), and it contained 85% of the biomass burning tracer compounds, 75% of the PAH and quinones, and less than 10% of the other tracer compounds, showing that biomass burning emission is an important source for toxic PAHs. The third and fourth components were clearly SOA related. The Bio SOA (Figure 6c) component contained 87% of the isoprene and $\alpha$-pinene oxidation products, and some other aged SOA compounds, such as SA (29%). The other SOA component (Figure 6d) contained the majority of dicarboxylic acids (SA, GLA, and AZA (40%), but also CPA (60%), which is a first generation oxidation product from $\alpha$-pinene.

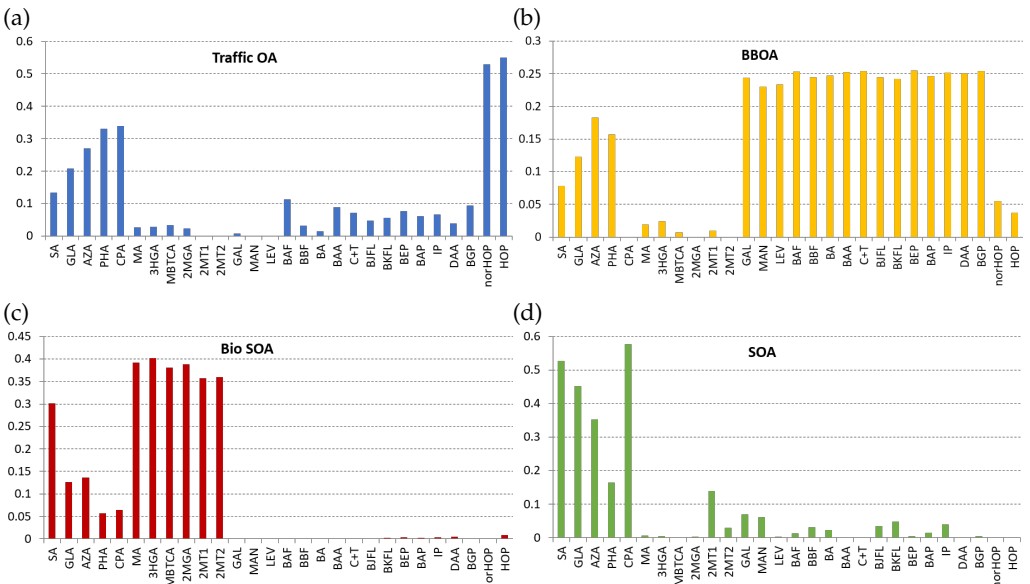

**Figure 6.** Loadings of the organic tracer compounds in the (**a**) Traffic OA; (**b**) BBOA; (**c**) Bio SOA; (**d**) SOA components, obtained after MCR-ALS analysis.

Figure 7a–c shows the average contribution of the four components to OA in Barcelona (Figure 7a), Manlleu (Figure 7b), and Bellver (Figure 7c), while the individual score values of the components in the analyzed $PM_{10}$ samples are given in Figure 7d. The figures show that the OA in the traffic station of Barcelona was dominated by traffic emissions (70%). The OA in the sub-urban background station of Manlleu was dominated by biomass burning emission (67%), and the OA in the rural background station of Bellver was dominated by SOA (47%). The Traffic OA score values strongly correlated with the mean $NO_2$ concentrations of the studied days ($R^2 = 0.7$; $p < 0.01$), pointing to hopanes as a good indicator for Traffic OA. The summer samples (02/08) are dominated by bio SOA in all stations. The high $O_3$ concentrations in summer may be related to elevated bio SOA concentrations, but there was only one summer sample to check this relationship. Further, 22/12 was the other sample day with high $O_3$ concentration, although this was the result of mixing of local air and air from the free troposphere. The Bio SOA was not high on this day. Bio SOA is probably mainly related to warmer summer months when the biogenic VOCs concentrations are also elevated, as it was observed in previous studies [9,16]. The samples of days 20/11 and 18/12, which were characterized by stagnant weather conditions, showed the highest score values for the local primary emission sources (i.e., traffic in Barcelona, and biomass burning in Manlleu and Bellver). In former studies, the sum of the score values of the OA components was correlated with organic carbon (OC) concentration [22], but OC

concentrations were not available in the present study to confirm this result. On the hand, the sum of scores showed a moderate correlations with $PM_{10}$ ($R^2$ = 0.3; $p < 0.05$). However, a lack of a stronger correlation with $PM_{10}$ was due to the contribution of inorganic material, such as the influenced by Saharan dust intrusion on 18/12. Accordingly, $PM_{10}$ events due to Saharan dust intrusion do not necessarily lead to proportionally higher OC and organic molecular tracer concentrations, which is in agreement with previous observations in the urban area of Barcelona [59]. In the present study, the dust intrusion on 18/12 resulted in very high $PM_{10}$ concentrations, which was only partially reflected by the higher OA score values on that day in the stations (Figure 7d).

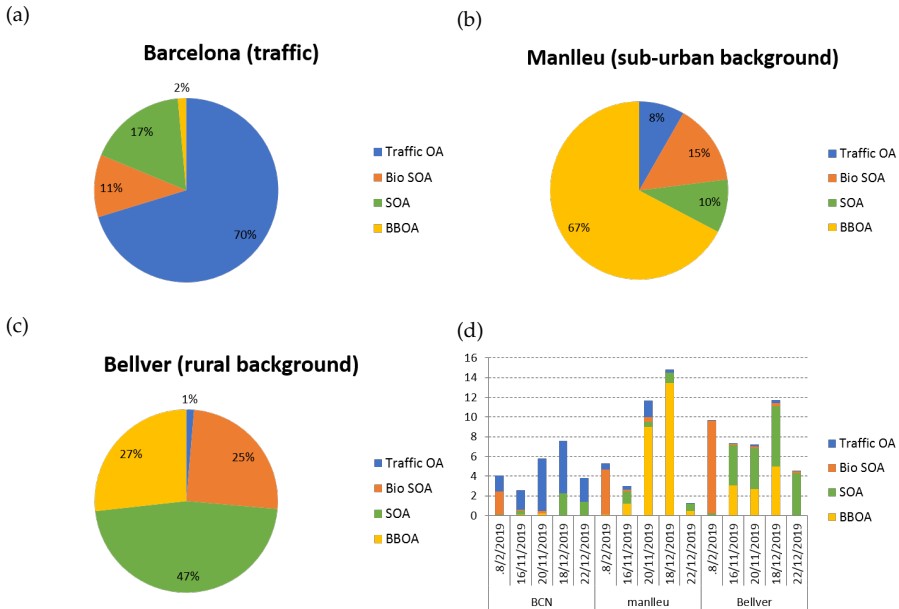

**Figure 7.** Contributions (**a**–**c**) and score values (**d**) of the four components obtained after MCR-ALS analysis.

### 3.5. PM Toxicity

The toxic effect of PM extracts was studied with lung epithelial cell lines A549 by detecting the cell viability in 24 h after exposure to different concentrations of the $PM_{10}$ extracts (expressed as $m^3$ of equivalent air volumes/mL) relative to the control (Figure 8). The results evidenced the strongest decrease of the cell viability, and thus higher toxic effects, in the $PM_{10}$ extracts from the sub-urban (Manlleu) and rural background (Bellver) stations of 20/11 and 18/12. These samples exhibited IC50 values lower than 10 $m^3$ eq.air/mL, and the Manlleu samples were slightly more toxic than the Bellver samples, with IC50s of 7.7 and 4.3 $m^3$ eq.air/mL, respectively, for 20/11 and 18/12 in Manlleu, compared to 9.6 and 6.3 $m^3$ eq.air/mL in Bellver. These days also exhibited stagnant atmospheric conditions and low PBLH. Generally, cells exposed to $PM_{10}$ extracts of the urban traffic station of Barcelona were more viable than those exposed to $PM_{10}$ extracts from Manlleu and Bellver de Cerdanya. An exception was observed for the 22/12 samples, in which the Barcelona extracts exhibited relatively higher cytotoxicity, with an IC50 equal to 19.1 $m^3$ eq.air/mL. $PM_{10}$ samples of 02/08 and 16/11 presented also low cytotoxicity. In 02/08, Manlleu samples induced more cell mortality than Bellver and Barcelona (IC50 = 14.4 $m^3$ eq.air/mL), while Manlleu and Bellver samples collected on 16/11 presented similar cytotoxicity (IC50s of 17.5 and 15.7 $m^3$ eq.air/mL, respectively).

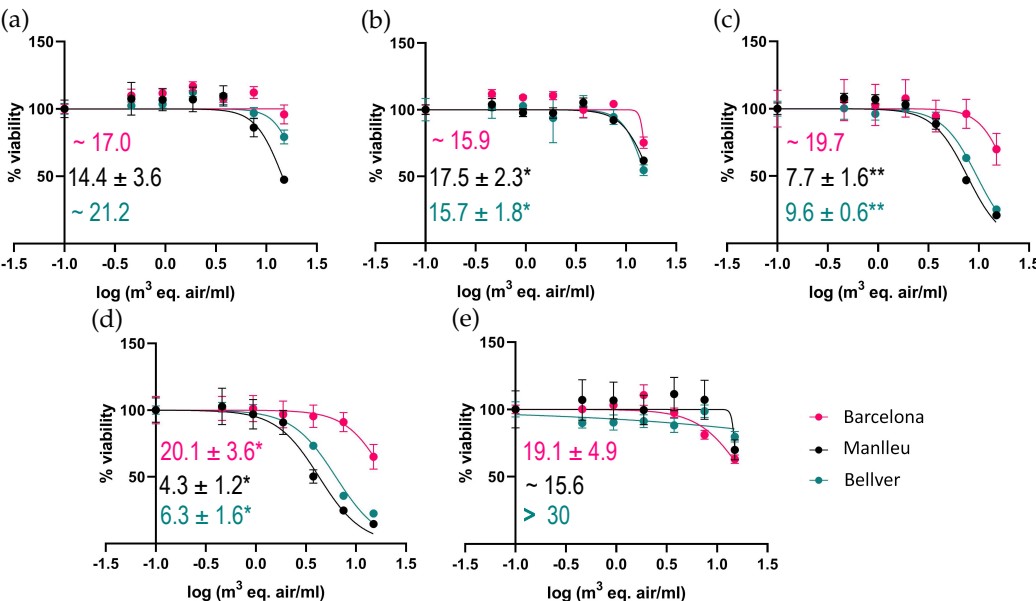

**Figure 8.** Toxicity curves and IC50 values (in m³ eq./mL) for each day and station. ∼ indicates approximate results, very wide IC50 slopes. * indicates slope fitting parameter $R^2 > 0.96$. ** indicates slope fitting parameter $R^2 > 0.99$ (**a**) 02/08/2019, (**b**) 16/11/2019, (**c**) 20/11/2019, (**d**) 18/12/2019, (**e**) 22/12/2019.

After the inspection of the four MCR-ALS components loadings and the cytotoxicity results, some relationships can be observed, such as the one between the biomass burning OA score values and IC50 ($R^2 = 0.6$; $p < 0.01$). Indeed, the most toxic samples from the sub-urban station (Manlleu: 20/11 and 18/12) were also those with the highest amounts of OA, mainly from biomass burning, and the highest concentrations of PAHs and quinones, followed by the samples from the rural background station (Bellver) on the same days. PAHs, TEQ-PAH, and quinone concentrations showed substantial correlations with IC50 ($R^2 > 0.5$; $p < 0.01$) in the analyzed samples. Other compound concentrations that corrrelated with IC50 were levoglucosan, mannosan, galactosan ($R^2 > 0.4$; $p < 0.01$), and azealic acid ($R^2 = 0.3$; $p < 0.05$), as the only compounds that could be linked to SOA. This suggests that the presence of compounds related to biomass burning emissions were responsible for triggering molecular mechanisms leading to lung cell death. This may be caused by PAHs and quinones, which have been linked to the alteration of cell development and functions, and eventually to cell death in former studies [60] as well as by other elemental content that have not been analyzed in this study [61]. The rural background (Bellver) samples of 20/11 and 18/12 also contained considerable amounts of SOA. The direct involvement of these compounds in cell death observed in these samples is doubtful in the present study since the 22/12 sample, whose composition is almost all SOA, has no toxic effect to cells. Further, the summer sample of 2/8 that was dominated by biogenic SOA showed very low toxic effects in all stations. However, in the urban traffic station (Barcelona), the most toxic samples (18–12 and 22/12) were those that had a significant contribution of SOA combined with traffic related emission. One possibility is that the combination of SOA and combustion emissions results in a synergistic behavior that enhances the intrinsic toxicity of both aerosols types [62,63]. The toxicity of the PM extracts obtained in this study is very relevant, since the IC50 equivalent air volumes between 4.4 m³ and 30 m³ are within the range of the daily respiration volumes in humans. The results show the high sensitivity of the cytotocxicity test with the lung epithelial cell line A549. On the other hand, it shows that the current levels of air pollutants may provoke health effects in the population.

### 3.6. Health Impact of Air Pollution in Catalonia

In order to gain insight into the health impact of air pollution in Catalonia, the annual mean concentrations of $PM_{10}$, $NO_2$, $O_3$, and BAP (Figure 2a–d) were divided by the air quality standards of the World Health Organization (WHO). The WHO annual limit values are 20 μg/m$^3$ for $PM_{10}$, 40 μg/m$^3$ for $NO_2$, whereas $O_3$ has a daily 8-h mean limit value of 100 μg/m$^3$. Regarding BAP, the WHO sets the acceptable risk value of BAP concentrations in $PM_{10}$ at 0.12 ng/m$^3$ based on recent epidemiological studies. BAP and other PAHs have often been evaluated on their carcinogenic effect, although there is recent evidence that PAHs at lower concentrations have other effects as well, such as impact on cognitive development in children [64,65]. The WHO limit values are lower than those of the EU directives. In the present study, the health risk of air pollution was evaluated with the WHO values, since the stations are situated near hospitals and schools, which lead to a more realistic situation of their actual health impact than the evaluation with EU standards. Moreover, the three selected stations often show high levels of air quality indicators. So, an acceptable air quality in these stations will probably also lead to acceptable air quality in other areas, while the EU standards may underestimate real epidemiological impacts of air pollution. Ratios of the annual average concentrations of $PM_{10}$, $NO_2$, $O_3$ and BAP (Figure 2) by the WHO air quality standard values > 1 indicate that the population is exposed to unacceptable air pollution levels. For the calculation of the overall health risk, the square root of sum of the squares for each indicator ratio was used (Figure 9d).

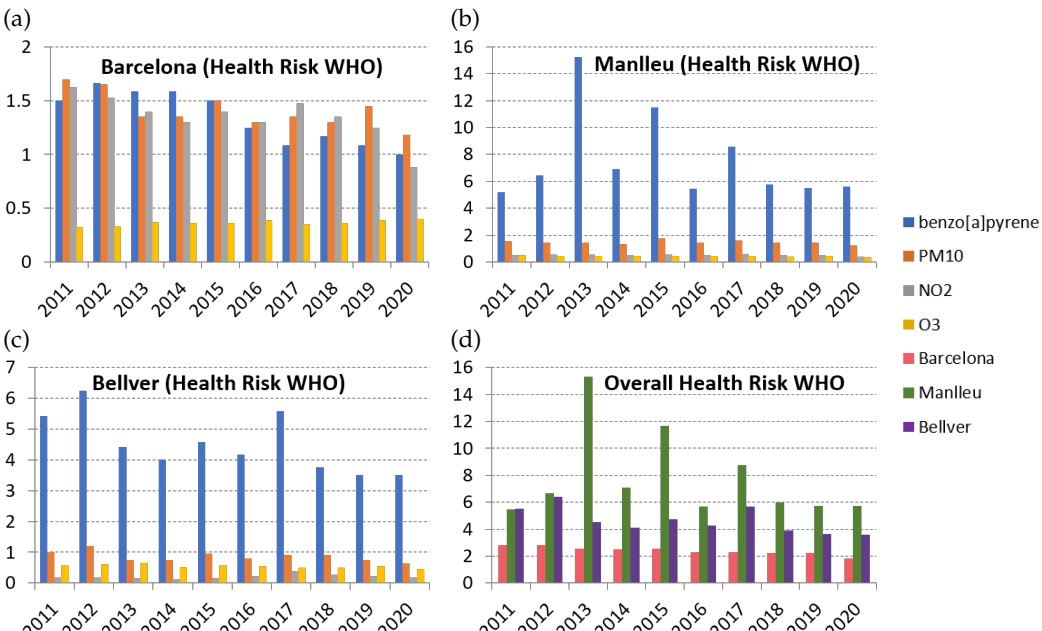

**Figure 9.** Health risk analysis of air quality in (**a**) Barcelona, (**b**) Manlleu, and (**c**) Bellver de Cerdanya based on the ratio of annual average $PM_{10}$, $NO_2$, $O_3$, and BAP concentrations versus the WHO air quality standards. Ratio > 1 is an unacceptable health risk. (**d**) Overall health risk is the square root of the sum of squares of the four air quality ratios in the three air quality network stations.

In the traffic station in Barcelona (Figure 9a), $PM_{10}$, $NO_2$, and BAP had an unacceptable health risk to its population. The overall health risk of air quality was >1, and the main emission source was traffic (Figure 7a). However, in 2020 $NO_2$ levels resulted in health risks <1 presumably due to the significant reduction of traffic in the city under severe COVID-19 lockdown during March and April [43]. In the sub-urban background station in Manlleu (Figure 9b), $PM_{10}$ and BAP had an unacceptable health risk to its population. The overall health risk to air pollution was also >1, which is mainly related to biomass burning emissions as observed in the source apportionment analysis (Figure 7b). In the rural background station of Bellver de Cerdanya (Figure 9c), BAP had an unacceptable

health risk to its population, while $PM_{10}$ was >1 in 2011 and 2012, and <1 in the other years. The overall health risk of air pollution in Bellver was unacceptable, and was also related to biomass burning emissions according to the source apportionment analysis (Figure 7c).

The health risk evaluation for air pollution in these three stations showed that primary emission from human related activities, i.e., traffic emissions in Barcelona, and biomass burning emissions in the sub-urban and rural background, were responsible for the air pollution in these sites, which was exacerbated under temperature inversion conditions, and leading to toxic effects.

Gaseous $O_3$ concentrations were not considered as an important health risk to the population in the studied air quality stations, although it has been indicated as a risk factor in the areas of Manlleu and Cerdanya in previous studies [19,38]. $O_3$ can reach high concentrations during a few hours, especially in summer, however, its levels drop rapidly due to its reactivity, which in turn can imply the formation of SOA and enhance PM concentrations. Therefore, the maximum daily 8-h average may be more appropriate for use, instead of the annual 24-h average concentration, as we chose to do in this study. In fact, in 2019, the maximum daily 8-h WHO standard of 100 µg/m$^3$ was exceeded on 66, 61, and 4 days in the air quality stations in Manlleu, Bellver de Cerdanya and Barcelona, respectively, while only 25 days are permitted in the air quality directive. The exceedance of 8-h $O_3$ concentrations was mainly occurring during summer, when also the highest bio SOA concentrations were measured. Both biogenic VOCs from these rural areas as well as anthropogenic VOCs and $NO_2$ from urban areas are involved in $O_3$ formation, so a reduction of the anthropogenic emissions can lead in turn to a reduction of the $O_3$ concentrations in sub-urban and rural background air. This may also have an effect on SOA formation in these areas.

A major limitation of the applied health risk assessment in the present study is the use of annual mean concentrations of the air quality indicators. First of all, a year needs to pass in order to obtain the levels of these indicators, so the evaluation is always retrospective. Second, the use of mean values is useful to evaluate chronic exposure, but it does not take into account the short-term effects of air pollution during peak events exposure [1,2]. Third, the results of this study and former ones have shown that the chemical PM composition is highly variable in space and time. $PM_{10}$ and BAP concentrations are influenced by multiple emission sources, such as traffic in urban areas and biomass burning in more rural areas, and atmospheric processes, such as secondary aerosol formation, as well as atmospheric conditions. Moreover, the levels of gaseous pollutants, i.e., $NO_2$ and $O_3$, interact in these processes. The use of organic tracer compounds have shown to be useful to distinguish aerosols from primary emission sources as well as secondary aerosols, to an extend that some of them, such as levoglucosan and methylterols, show the presence of biomass burning organic aerosols and biogenic secondary organic aerosol, respectively. This information is helpful in order to understand elevated PM concentration in sub-urban and rural sites, where traffic and industrial emissions are less important. A clear improvement of the health risk assessment are obtained with models that predict the air quality, so that measurements can be taken to avoid or limit the air pollution exposure to humans and the environment. Different approaches exists to forecast the air quality, such as deterministic, statistical, or neural network models [21,66–68]. Generally, these models predict within an acceptable error the presence and levels of $PM_{10}$, $NO_2$, and $O_3$. Nevertheless, in the case of the studied area of Catalonia, the forecast models [21,68] were less accurate for $PM_{10}$, and often underestimated the measured values, due to a lack of information of SOA formation and biomass burning emissions, among other factors. The findings of the present study can be useful to improve these models and limit the health risk to air pollution exposure.

## 4. Conclusions

Air quality indicators, i.e., $PM_{10}$, $NO_2$, $O_3$ and benzo[a]pyrene, have been evaluated in three geographically different sites in Catalonia (Spain); an urban traffic station in

Barcelona, a sub-urban background station in Manlleu, and a rural background station in Bellver de Cerdanya, over the past decade. Organic tracer compounds in PM$_{10}$ and toxicity were analyzed between summer and winter 2019 in selected samples from these stations under contrasting atmospheric conditions and air pollution load.

Health risk assessment for chronic exposure, using WHO air quality standards, showed that NO$_2$, PM$_{10}$ and benzo[a]pyrene from traffic emissions posed an unacceptable risk to the human population in the urban traffic site in Barcelona; PM$_{10}$ and benzo[a]pyrene from biomass burning were unacceptably high in the sub-urban site in Manlleu; and benzo[a]pyrene from biomass burning was unacceptably high in the rural background site of Bellver de Cerdanya. Although O$_3$ did not show unacceptable health risks for chronic exposure, on a short term eight-hour base, the O$_3$ concentration standard was exceeded >25 days in 2019 in Manlleu and Bellver de Cerdanya, indicating potentially acute health effects in the population of the sub-urban and rural areas for this contaminant. This approach was limited to chronic exposure and does not include shor-term exposure to peak concentrations of air pollution that are often present in the studied area.

These results request different mitigation strategies for urban and rural sites in order to improve the air quality. In urban areas, traffic emissions are still the dominating factor that influence the air quality and may directly be responsible for part of the SOA and O$_3$ levels in sub-urban and rural areas. The sub-urban and rural areas cope with air pollution from local biomass burning emissions. Biomass burning in the studied sub-urban and rural sites was related to the presence of forests and agricultural fields. Moreover, biomass burning was also highly related to the period between October and March, due to permissions for open field fires in this period, as well as the use of wood for domestic heating in this period of cold weather. Despite its natural origin, incomplete combustion of biomass generates large quantities of toxic compounds. In the near future, regulations and vehicle replacement for lower emission devices will hopefully improve the air quality in sites that are affected by traffic emissions. On the other hand, biomass burning is a more complex issue to tackle when multiple small scale sources are involved, such as combustion installations for domestic heating and vegetative waste removal in open fires in agricultural fields. Notwithstanding, control systems for these types of sources, and regulation of combustion efficiencies of installations are necessary. This regulation should not involve only PM$_{10}$, but also PAH emissions, since PAH is a better risk indicator for combustion than PM$_{10}$. Representative toxicity tests in the present study and former ones [60,61] showed that biomass smoke affects the general cellular functions of organisms and can lead to cell death at relevant air volumes. The toxicity of primary emissions is relatively well-understood but there is a lack of information on the toxic effects of SOA. The present study showed that SOA does not contribute to PM toxicity compared to, for example, biomass burning. Nevertheless, recent studies showed that urban SOA induced moderate effects in exposure assays [62,63,69,70]. On the other hand, risk assessment studies are in agreement with the main results of the present study, showing that ambient air pollution from biomass burning in rural–sub-urban sites represents a higher risk to develop lung cancer in humans throughout their life than pollutants that were measured in modern cities, such as Barcelona, with traffic as a dominant emission source [71]. The results of the present study can be useful to improve models that forecast air pollution in the region and other areas, so that the exposure to air pollution in human and the environment can be avoided or limited.

**Author Contributions:** Conceptualization, B.L.v.D.; formal analysis, C.J., P.V., B.L.v.D. and C.B.; investigation, B.L.v.D.; data curation, C.J., P.V., M.U., C.B. and B.L.v.D., writing—original draft preparation, C.J. and B.L.v.D.; writing—review and editing, all authors; supervision, B.L.v.D., funding acquisition P.F., J.O.G. and C.B. All authors have read and agreed to the published version of the manuscript.

**Funding:** This work was supported by the Spanish Ministry of Science and Innovation (INTEMPOL PGC2018-10228-B-I00) and the IDAEA-CSIC Centre of Excellence Severo Ochoa Project CEX2018-000794-S. C.J. also acknowledges financial support from the Spanish Ministry of Universities (FPU19/06826).

**Institutional Review Board Statement:** Not applicable.

**Informed Consent Statement:** Not applicable.

**Data Availability Statement:** Not applicable.

**Acknowledgments:** Technical assistance from Roser Chaler and Alexandre Garcia is acknowledged. The authors thank the Public Health Agency of Barcelona and the Department of Territory and Sustainability of the Catalan Government for the supply of $PM_{10}$ samples and the valuable discussion and critical reading of the manuscript.

**Conflicts of Interest:** The authors declare no conflict of interest.

## Appendix A. Complementary Figures

This appendix provides complementary figures of the multi-Variant Regression Curve—Alternating Least Square (MCR-ALS) analysis applied for filter selection described in Section 2.1.

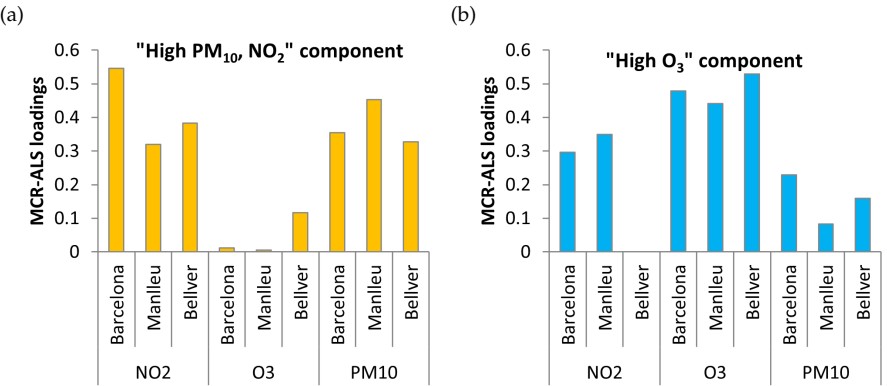

**Figure A1.** Loadings of the two components from the MCR-ALS analysis on the $PM_{10}$, $NO_2$, and $O_3$ data from Barcelona, Manlleu and Bellver de Cerdanya. (**a**) "High $PM_{10}$, $NO_2$" component. (**b**) "High $O_3$" component.

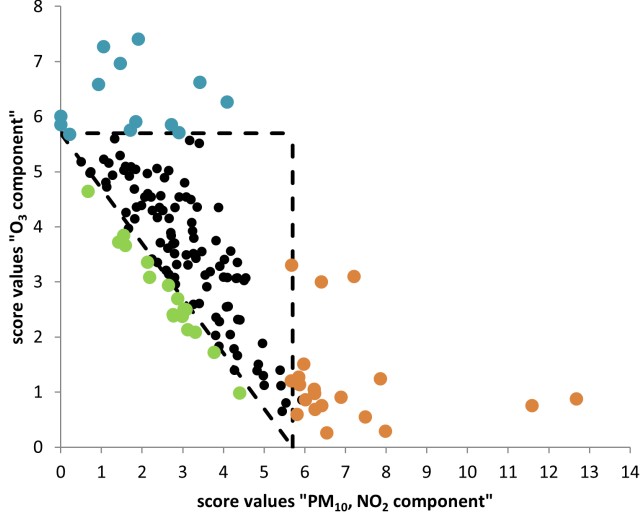

**Figure A2.** Score values of the two components from the MCR-ALS analysis on the $PM_{10}$, $NO_2$, and $O_3$ data from Barcelona, Manlleu and Bellver de Cerdanya between august and December 2019. The 5.6 cut-off value is indicated for the selection of days, for: (1) high $PM_{10}$ and $NO_2$ concentrations in the tree stations (Orange), (2) high $O_3$ concentrations in all three stations (Blue), and (3) low $PM_{10}$ in all three stations (Green).

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
