# Peer review of "Source Apportionment and Toxicity of PM in Urban, Sub-Urban, and Rural Air Quality Network Stations in Catalonia"

_atmosphere, doi:10.3390/atmos12060744_

Round 1

Reviewer 1 Report

This research is very interesting. The author discuss  apportionment and toxicity of PM in urban, suburban, and rural air quality network stations, but some suggestions need to be a concern.

  1. The literature review should be extended. Especially in the literature review. The authors should mention the previous study about the impact, transition, and duration frequency of PM [1]-[5].
  2. 2. Section 2.1 needs to explain how the author gets the data from pre-processing and processing. Especially if there are missing values. How you handle this?
  3. 3. The statistical method chosen is very unclear and on line 259 the author suddenly mentions the value of R2 without explaining what he wants to know. What types of correlation do authors use? Although in the appendix. The author explains that using multi-variant Regression Alternating Least Square. A complete description of this model is needed!
  4. 4. Check also the classical assumptions that need to be used by regression analysis such as Linear relationship, Multivariate normality, No or little multicollinearity, No auto-correlation, and Homoscedasticity [6]

Additional reference:

[1]      N. A. H. Janssen, P. Fischer, M. Marra, C. Ameling, and F. R. Cassee, “Short-term effects of PM 2.5 , PM 10 and PM 2.5-10 on daily mortality in the Netherlands,” Sci. Total Environ., 2013, doi: 10.1016/j.scitotenv.2013.05.062.

[2]      N. Masseran and M. A. M. Safari, “Modeling the transition behaviors of PM 10 pollution index,” Environ. Monit. Assess., vol. 192, no. 441, pp. 1–15, 2020.

[3]      R. E. Caraka, R. C. Chen, H. Yasin, B. Pardamean, T. Toharudin, and S. H. Wu, “Prediction of Status Particulate Matter 2.5 using State Markov Chain Stochastic Process and HYBRID VAR-NN-PSO,” IEEE Access, vol. 7, pp. 161654–161665, 2019, doi: 10.1109/ACCESS.2019.2950439.

[4]      R. E. Caraka, R. C. Chen, H. Yasin, Y. Lee, and B. Pardamean, “Hybrid Vector Autoregression Feedforward Neural Network with Genetic Algorithm Model for Forecasting Space-Time Pollution Data,” Indones. J. Sci. Technol., vol. 6, pp. 243–266, 2021.

[5]      N. Masseran and M. A. Mohd Safari, “Intensity–duration–frequency approach for risk assessment of air pollution events,” J. Environ. Manage., vol. 264, p. 110429, 2020, doi: 10.1016/j.jenvman.2020.110429.

[6]      R. H. Myers, D. C. Montgomery, G. G. Vining, and T. J. Robinson, Generalized Linear Models: With Applications in Engineering and the Sciences: Second Edition. 2012.

Reviewer 2 Report

Regarding the manuscript entitled "Source apportionment and toxicity of PM in urban, suburban and rural air quality network stations in Catalonia". The study presents some interesting results, but the data used are extremely limited to reach any safe conclusions. As already mentioned, the main limitation of the study is the low number of samples used and additionally the lack of discussion of some decisions from the authors. My other two major concerns are the source apportionment implementation and the use of trajectories. SA technique is not described at all, and additionally, the number of samples used is extremely low especially considering that the number of tracers is really higher. Secondly, the trajectories are estimated for a very low height and the surface effect may highly increase the uncertainty of the results. It is also a fact that single trajectories are not enough to provide a good idea for the spatial origin/variation and statistical trajectory methods should be used, which in this case is impossible given the low number of samples. I also find it very strange that results are presented in the methodology section.  Overall, I find the study not suitable for publication unless more results are included.

Reviewer 3 Report

This paper studies the sources and toxicity of PM in three different regions of Catalonia. I think the study is interesting. But there are some limitations in this study. One main concern is the conclusions are only based on five samples from each sampling site, which is rare, although the authors have explained how they chose these five samples. For example, based on their results, the authors conclude SOA does not contribute to PM toxicity. According to some previous studies, this may not be true (See Chowdhury et al., es&t letters 2018, Fushimi et al., Sci Tot Environ 2021). Another limitation is that the source apportionment results are estimated from five days samples (in total 15 samples). The results from such few samples are not very reliable. In addition, the toxicity results are not well discussed with the chemical composition results.

Minor comments:

  1. Line 78: change to A549.
  2. The language can be further improved.

Round 2

Reviewer 1 Report

  1. Authors do not update references as informed, especially in the literature review. The authors should mention the previous study about the impact, transition, and duration frequency of PM [1]-[5]. These three points need to be mentioned because they are very important, because this is the topic the author is discussing.
  2. Still, the author should check also the classical assumptions that need to be used by regression analysis such as Linear relationship, Multivariate normality, No or little multicollinearity, No auto-correlation, and Homoscedasticity. Or at least, authors should mention the limitation of this research, that you are not performing regression assumptions test.

I give a second chance to author revise the manuscript. Otherwise, this paper deserves to be rejected because the analysis step is still wrong and ignores statistical parametric tests.

Additional reference:

[1]      N. A. H. Janssen, P. Fischer, M. Marra, C. Ameling, and F. R. Cassee, “Short-term effects of PM 2.5 , PM 10 and PM 2.5-10 on daily mortality in the Netherlands,” Sci. Total Environ., 2013, doi: 10.1016/j.scitotenv.2013.05.062.

[2]      N. Masseran and M. A. M. Safari, “Modeling the transition behaviors of PM 10 pollution index,” Environ. Monit. Assess., vol. 192, no. 441, pp. 1–15, 2020.

[3]      R. E. Caraka, R. C. Chen, H. Yasin, B. Pardamean, T. Toharudin, and S. H. Wu, “Prediction of Status Particulate Matter 2.5 using State Markov Chain Stochastic Process and HYBRID VAR-NN-PSO,” IEEE Access, vol. 7, pp. 161654–161665, 2019, doi: 10.1109/ACCESS.2019.2950439.

[4]      R. E. Caraka, R. C. Chen, H. Yasin, Y. Lee, and B. Pardamean, “Hybrid Vector Autoregression Feedforward Neural Network with Genetic Algorithm Model for Forecasting Space-Time Pollution Data,” Indones. J. Sci. Technol., vol. 6, pp. 243–266, 2021.

[5]      N. Masseran and M. A. Mohd Safari, “Intensity–duration–frequency approach for risk assessment of air pollution events,” J. Environ. Manage., vol. 264, p. 110429, 2020, doi: 10.1016/j.jenvman.2020.110429.

Reviewer 2 Report

The authors addressed the points raised by reviewers adequately and the manuscript is noe=w at a state that can be published.

Author Response

A complete revision of the text was done to improve the English language of the manuscript. Changes in the manuscript have been introduced into another version in LaTex format. This revised version, and a PDF with the changes compared to the former version, is sent to the journal.

Reviewer 3 Report

The manuscript can be accepted

Author Response

(The authors gave the same response as above.)
